# Distributional Convergence of the Sliced Wasserstein Process

Jiaqi Xi[1] and Jonathan Niles-Weed[1,2]

[1]Courant Institute of Mathematical Sciences, New York University, NY 10012
[2]Center for Data Science, New York University, NY 10011

## Abstract

Motivated by the statistical and computational challenges of computing Wasserstein distances in high-dimensional contexts, machine learning researchers have defined modified Wasserstein distances based on computing distances between one-dimensional projections of the measures. Different choices of how to aggregate these projected distances (averaging, random sampling, maximizing) give rise to different distances, requiring different statistical analyses. We define the *Sliced Wasserstein Process*, a stochastic process defined by the empirical Wasserstein distance between projections of empirical probability measures to all one-dimensional subspaces, and prove a uniform distributional limit theorem for this process. As a result, we obtain a unified framework in which to prove sample complexity and distributional limit results for all Wasserstein distances based on one-dimensional projections. We illustrate these results on a number of examples where no distributional limits were previously known.

## 1 Introduction

The Wasserstein distances have become useful tools in machine learning and data science, with applications in transfer learning [6, 33], generative modeling [4, 19], statistics [8, 20], and various scientific domains [36, 44]. Despite the popularity of these distances, they suffer from serious drawbacks in high dimensions. From a statistical standpoint, estimating the Wasserstein distances from data suffers from the *curse of dimensionality*, with convergence rates degrading sharply as the dimension increases [16, 29, 37, 43]. From a computational standpoint, despite recent algorithmic advances [1, 7], the best algorithms for approximately computing general Wasserstein distances between distributions supported on $n$ points in $\mathbb{R}^d$ for $d \geq 2$ have running times scaling quadratically in $n$, which is prohibitive on very large data sets. These deficiencies have motivated the development of modifications of the Wasserstein distances which reduce the high-dimensional case to a series of one-dimensional problems.

Given two compactly supported probability distributions $P$ and $Q$ in $\mathbb{R}^d$, we write $P_u$ and $Q_u$ for the projections of $P$ and $Q$ onto the one-dimensional subspace spanned by $u$, for any $u \in \mathbb{S}^{d-1}$. Explicitly, if $X \sim P$, we let $P_u$ denote the law of $u^\top X$. The measures $P_u$ and $Q_u$ are probability distributions on $\mathbb{R}$ obtained by collapsing $P$ and $Q$ to the one-dimensional "slice" in the direction of $u$. Crucially, no matter how large $d$ is, the Wasserstein distance $W_p^p(P_u, Q_u)$ between the one dimensional measures is always easy to work with: it can be estimated from data at the rate that is independent of the dimension, and if $P_u$ and $Q_u$ are supported on $n$ points, then $W_p^p(P_u, Q_u)$ can be computed in *nearly linear* time by a simple sorting procedure.

This observation has given rise to a number of different proposals for defining a distance between $P$ and $Q$ by aggregating the one-dimensional distances, the most prominent of which is the sliced

Wasserstein distance [3, 32]:

$$SW_p^p(P,Q) := \int W_p^p(P_u, Q_u)\, d\sigma(u)\,, \tag{1}$$

where $\sigma$ denotes the uniform measure on $\mathbb{S}^{d-1}$. Other options include:
- Discrete Sliced Wasserstein distance: $\widehat{SW}_p^p(P,Q) := \frac{1}{L}\sum_{i=1}^L W_p^p(P_{u_i}, Q_{u_i})$, $\{u_i\} \subseteq \mathbb{S}^{d-1}$ [3].
- Max-Sliced Wasserstein distance: $MSW_p^p(P,Q) := \max_{u\in\mathbb{S}^{d-1}} W_p^p(P_u, Q_u)$ [15, 29, 31].
- Distributional Sliced Wasserstein distance: $DSW_p^p(P,Q) := \sup_{\tau\in\mathcal{P}_C} \int W_p^p(P_u, Q_u)\, d\tau(u)$, where $\mathcal{P}_C$ is a subset of probability measures on $\mathbb{S}^{d-1}$ [28].

Though the details of these techniques differ, they can be put on a common footing: if we view the function $W : u \mapsto W_p^p(P_u, Q_u)$ as a bounded function on $\mathbb{S}^{d-1}$, then each of these sliced distances takes the form $F(W)$ for some function $F : \ell^\infty(\mathbb{S}^{d-1}) \to \mathbb{R}$.

Since these distances are all based on one-dimensional projections, it is natural to conjecture that they enjoy improved statistical performance. This conjecture has been verified in certain special cases [25, 27, 29] However, the analysis of these distances has largely been conducted separately, with different arguments tailored to each distance. This raises the following fundamental question: is there a *unified* approach to the analysis of these distances, which provides statistical guarantees for all of them simultaneously?

In this work, we develop such a unified approach. In addition to generalizing prior works, our techniques allow us to prove new distributional convergence results for the sliced Wasserstein distance and its many variants. These results make it possible to construct asymptotically valid confidence intervals for variants of the sliced Wasserstein distances and to guarantee the validity of the bootstrap. Prior to our work, such results were only known for the standard sliced Wasserstein distance (1) [25] or for the sliced and max-sliced Wasserstein distances between discrete distributions [30].[1]

Obtaining distributional limits for empirical Wasserstein distances is an active area of research. In the one-dimensional case, fundamental contributions were made by [10, 11, 12], and further progress has been made in the case where one or both of the measures are discrete [14, 38, 40]. Multi-dimensional limits were recently obtained by [9, 13], but these are not centered at the population-level quantities, making them of limited utility for inference. However, when the distributions are very smooth, there exist estimators with distributional limits with good centering [26]. In this work, we draw on techniques recently proposed in [22] to obtain central limit theorems by exploiting duality.

We consider compactly supported probability measures $P$ and $Q$ in $\mathbb{R}^d$ with connected supports, and the Wasserstein distances $W_p^p$ for $p > 1$. To analyze the empirical behavior of the sliced Wasserstein distance and its variants, we define a stochastic process

$$\mathbb{G}_n(u) := \sqrt{n}(W_p^p(P_{nu}, Q_{nu}) - W_p^p(P_u, Q_u)) \qquad u \in \mathbb{S}^{d-1}\,. \tag{2}$$

where $P_n$ and $Q_n$ consist of i.i.d. samples. We may view $\mathbb{G}_n$ as a random element of $\ell^\infty(\mathbb{S}^{d-1})$, which records the deviation of the Wasserstein distance from its population counterpart along every direction simultaneously. We call $\mathbb{G}_n$ the *Sliced Wasserstein Process*.

Our main result shows that

$$\mathbb{G}_n \rightsquigarrow \mathbb{G} \in \ell^\infty(\mathbb{S}^{d-1})\,, \tag{3}$$

where $\mathbb{G}$ is a tight Gaussian process on $\mathbb{S}^{d-1}$. That is, the collection of random variables $\sqrt{n}(W_p^p(P_{nu}, Q_{nu}) - W_p^p(P_u, Q_u))$ indexed by elements of $\mathbb{S}^{d-1}$ enjoys a *uniform* central limit theorem. As is well known in the statistics literature [41], uniform central limit theorems of this type give rise to distributional limits for any sufficiently regular functional on $\ell^\infty(\mathbb{S}^{d-1})$ via the functional delta method—in particular, we directly obtain distributional limit theorems for the sliced Wasserstein distance and its many variants as a special case. Our results likewise give techniques for proving the consistency of the bootstrap for any of the mentioned functionals as a consequence of general results for uniform central limit theorems.

---

[1]Concurrently and independently of our work, [21] proved distributional limits for the sliced and max-sliced Wasserstein, but not for other variants, as a byproduct of general results for distributional limits for Wasserstein distances.

## 2 Main Result

Throughout, $P$ and $Q$ denote two probability distributions in $\mathbb{R}^d$ with compact supports, contained in a closed ball $\overline{B(0,R)}$ around the origin. We fix a $p > 1$, and consider the Wasserstein distance of order $p$:

$$W_p^p(P,Q) = \inf_{\pi \in \Pi(P,Q)} \int \|x - y\|^p \, d\pi(x,y), \tag{4}$$

where the infimum is taken over all couplings between $P$ and $Q$. It is well known (see [42]) that this problem possesses a dual formulation:

$$W_p^p(P,Q) = \sup_{f:\overline{B(0,R)} \to \mathbb{R}} \int f \, dP + \int f^c \, dQ, \tag{5}$$

where $f^c$ denotes the $c$-transform: $f^c(y) = \inf_{x \in \overline{B(0,R)}} \|x - y\|^p - f(x)$. It can be shown (e.g.,[24, Lemma 1 & 5]) that the supremum in this dual formulation is achieved, and that without loss of generality we may assume that $f$ satisfies $f(0) = 0$ and $\|f\|_{\text{Lip}} \leq pR^{p-1}$. We denote the class of such functions by $\mathcal{C}$, and call any maximizer a *Kantorovich potential*. In order to obtain Gaussian limits, we adopt the following assumption:

(**CC**) For all $u \in \mathbb{S}^{d-1}$, the support of $P_u$ or $Q_u$ is an interval.

For $p > 1$, assumption (**CC**) guarantees that the supremum in (5) is achieved at a *unique* Kantorovich potential in $\mathcal{C}$ [35, Proposition 7.18]. In the absence of this uniqueness, Gaussian limits fail to hold for the optimal transport problem, even for discrete measures [38].

We now state our main result.

**Theorem 2.1.** *Suppose that $P$ and $Q$ are two probability distributions in $\mathbb{R}^d$ whose supports are contained in the closed $d$-ball $\overline{B(0,R)}$ for some $R > 0$. Assume that $P$ and $Q$ satisfy (**CC**). Let $P_n$ and $Q_m$ denote empirical measures consisting of $n$ and $m$ i.i.d. samples from $P$ and $Q$, respectively. If $n/(n+m) \to \lambda \in (0,1)$ as $n,m \to \infty$, then*

$$\sqrt{\frac{nm}{n+m}} \left( W_p^p(P_{n\cdot}, Q_{m\cdot}) - W_p^p(P_\cdot, Q_\cdot) \right) \rightsquigarrow \mathbb{G} \quad in \ \ell^\infty(\mathbb{S}^{d-1}), \tag{6}$$

*where $\mathbb{G}$ is a tight zero-mean Gaussian process on $\mathbb{S}^{d-1}$ with covariance function*

$$\mathbb{E}\mathbb{G}(u)\mathbb{G}(v) = (1-\lambda) \int f_u(u^\top x) f_v(v^\top x) \, dP(x) + \lambda \int f_u^c(u^\top y) f_v^c(v^\top y) \, dQ(y)$$
$$- (1-\lambda) \int f_u(u^\top x) \, dP(x) \int f_v(v^\top x) \, dP(x) \tag{7}$$
$$- \lambda \int f_u^c(u^\top y) \, dQ(x) \int f_v^c(v^\top Y) \, dQ(x),$$

*where $f_u, f_v \in \mathcal{C}$ are the unique Kantorovich potentials for $(P_u, Q_u)$ and $(P_v, Q_v)$, respectively.*

Theorem 2.1 formally includes the one-sample case as well, by taking $\lambda = 0, 1$.

**Remark 2.2.** *The assumption of compact support guarantees that the set of Kantorovich potentials corresponding to $P_u$ and $Q_u$ for any $u \in \mathbb{S}^{d-1}$ is uniformly Lipschitz, and is therefore a subset of a Donsker class. If the supports of $P$ and $Q$ were unbounded, in order to deduce the Donsker property, we would need additional assumptions on the 1-dimensional projections of $P$ and $Q$ as well as the cost function, (see e.g. [22, Theorem 5.2]) that do not hold for p-Wasserstein distances ($p > 1$) in general.*

**Remark 2.3.** *In the proof, assumption (**CC**) is only used to guarantee that for each $u \in \mathbb{S}^{d-1}$, there exists a unique Kantorovich potential achieving the supremum in the dual formulation of $W_p^p(P_u, Q_u)$. It is therefore possible to replace (**CC**) by any weaker assumption know to guarantee uniqueness [39, 45], but we adopt (**CC**) because it is the simplest such assumption we are aware of. In particular, Theorem 2.1 holds for $p = 1$ under the additional assumption that, for each $u \in \mathbb{S}^{d-1}$, the Kantorovich potential for $W_1(P_u, Q_u)$ is unique.*

As alluded to above, Theorem 2.1 gives rise to a wealth of statistical theorems as easy corollaries. To describe these implications, we return to the abstract setting described above: denote by $W : \mathbb{S}^{d-1} \to \mathbb{R}$ the function $W(u) = W_p^p(P_u, Q_u)$, and consider any functional $F : \ell^\infty(\mathbb{S}^{d-1}) \to \mathbb{R}$. Then the distances we consider take the form $F(W)$. By different choices of $F$, we obtain the sliced Wasserstein distance, the max-sliced Wasserstein distance, and the other variants described above. Theorem 2.1 will allow us to compare $F(W)$ to its empirical counterpart $F(W_{nm})$, where $W_{nm}(u) = W_p^p(P_{nu}, Q_{mu})$.

We recall the definition of directional Hadamard differentiability [34]: we say that $F$ is directionally Hadamard differentiable at $\Phi$ if for all sequences $h_n \searrow 0$ and $\Psi_n \to \Psi \in \ell^\infty(\mathbb{S}^{d-1})$, the limit

$$\lim_{n\to\infty} \frac{F(\Phi + h_n \Psi_n) - F(\Phi)}{h_n} =: F'_\Phi(\Psi)$$

exists. We verify in Section 3 the directional Hadamard differentiability of several examples. Under this assumption, we have the following.

**Corollary 2.4.** *Assume $F$ is directionally Hadamard differentiable. Under the assumptions of Theorem 2.1,*

$$\sqrt{\frac{nm}{n+m}}(F(W_{nm}) - F(W)) \rightsquigarrow F'_W(\mathbb{G}).$$

*Proof.* See [34]. $\qquad\square$

We also obtain a consistency result for the bootstrap, which we state for simplicity in the $n = m$ case.

**Corollary 2.5.** *Assume that $F$ is directionally Hadamard differentiable, and adopt the assumptions of Theorem 2.1. Let $P_n = \frac{1}{n}\sum_{i=1}^n \delta_{X_i}$ and $Q_n = \frac{1}{n}\sum_{i=1}^n \delta_{Y_i}$, and for $k \ll n$ denote by $P^*$ and $Q^*$ bootstrap empirical measures consisting of $k$ i.i.d. draws from $P_n$ and $Q_m$, respectively, and set $W^* = W_p^p(P_u^*, Q_u^*)$. If $k \to \infty$ and $k/n \to 0$, then*

$$\sup_{h\in BL(1)} \mathbb{E}[h(\sqrt{k}(F(W^*)) - F(W_n))|X_1, \ldots, X_n, Y_1, \ldots, Y_n]$$

$$- \mathbb{E}[h(\sqrt{n}(F(W_n) - F(W)))] \xrightarrow{p} 0,$$

*where $BL(1)$ is the set of functions with bounded Lipschitz norm 1.*

*Proof.* See [17]. $\qquad\square$

Finally, when the functional $F : \ell^\infty(\mathbb{S}^{d-1}) \to \mathbb{R}$ has a *linear* Hadamard derivative, the resulting statistic will again be asymptotically Gaussian. The following uniform convergence result shows that we can consistently estimate the covariance function of $\mathbb{G}$ from data, which can be used to obtain asymptotic confidence intervals in this setting.

**Theorem 2.6.** *Under the same assumptions as Theorem 2.1, there exists an estimator $\{\hat{\Sigma}_{u,v}\}_{u,v\in\mathbb{S}^{d-1}}$ for the covariance functions $\{\Sigma_{u,v}\}_{u,v\in\mathbb{S}^{d-1}}$ of the limiting process $\mathbb{G}$ in the sense that*

$$\mathbb{E}_{P,Q} \sup_{u,v\in\mathbb{S}^{d-1}} \|\hat{\Sigma}_{u,v} - \Sigma_{u,v}\|_\infty \to 0 \quad \text{as } n \to 0. \tag{8}$$

**Remark 2.7.** *Theorem 2.6 can be used to obtain asymptotic confidence intervals via Slutsky's theorem. For instance, if the functional $F$ has a linear derivative $F'_W$ which is of the form $F'_W(\Phi) = \int \Phi(u)\,d\tau(u)$ for a Borel measure $\tau$, then Theorem 2.6 implies that $\hat{\sigma}^2 := \int\int \hat{\Sigma}_{u,v}\,d\tau(u)\,d\tau(v)$ converges in probability to $\mathrm{var}(F'_W(\mathbb{G}))$, and therefore that $F(W_{nm}) \pm \hat{\sigma} z_{\delta/2}\sqrt{\frac{n+m}{nm}}$ is an asymptotic $(1-\delta)$ confidence interval for $F(W)$.*

## 3 Applications

In this section, we focus on three of the variants we discussed—the sliced, max-sliced, and distributional sliced Wasserstein distances—and show how our main results obtained in the previous section can be used to obtain accurate asymptotic inference for these quantities. For notational simplicity, we focus on the case where $n = m$, and rescale the resulting Gaussian process by a factor of $\sqrt{2}$.

## 3.1 Sliced Wasserstein Distance

Asymptotic and finite-sample inference for the sliced Wasserstein distance (SW) has already been thoroughly studied by [25]. We show that we can recover some of their results from our techniques. Their main focus was on a robustification of the SW distance, the "trimmed" SW distance, defined as

$$SW_{p,\delta}(P,Q) := \left( \int_{\mathbb{S}^{d-1}} \int_\delta^{1-\delta} |F_u^{-1}(t) - G_u^{-1}(t)|^p \, \mathrm{d}t \, \mathrm{d}\sigma(u) \right)^{\frac{1}{p}},$$

where $\sigma$ denotes the uniform probability measure on $\mathbb{S}^{d-1}$ and $F_u^{-1}$, $G_u^{-1}$ are the (pseudo-)inverses of the CDFs of $P_u$, $Q_u$ respectively. When $\delta = 0$, $SW_{p,\delta}$ reduces to the original sliced Wasserstein distance $SW_p$. The trimmed SW distance $SW_{p,\delta}$ discards the mass of $P_\theta$ and $Q_\theta$ above and below the $1 - \delta$ and $\delta$ quantiles (hence the term "trimmed"), and is therefore more robust in the face of outliers. This robustification is necessary when $P$ and $Q$ are no longer assumed to have compact supports. [25] derive Gaussian limits and bootstrap consistency for this functional.

To see how their results for the standard sliced Wasserstein distance (i.e., $\delta = 0$) can be derived under our stricter assumptions from Theorem 2.1, we denote by $F : \ell^\infty(\mathbb{S}^{d-1}) \to \mathbb{R}$ the integration functional:

$$F(\Phi) = \int \Phi(u) \, \mathrm{d}\sigma(u).$$

The dominated convergence theorem immediately implies that $F$ is Hadamard differentiable, with derivative

$$F'_\Phi(\Psi) = \int \Psi(u) \, \mathrm{d}\sigma(u).$$

We obtain the following.

**Theorem 3.1.** *Suppose that two compactly supported probability distributions $P$ and $Q$ in $\mathbb{R}^d$ satisfy (CC). Then*

$$\sqrt{n}(SW_p^p(P_n, Q_n) - SW_p^p(P, Q)) \xrightarrow{d} S := \int_{\mathbb{S}^{d-1}} \mathbb{G}(\theta) \, \mathrm{d}\sigma^d(\theta). \tag{9}$$

The random variable $S$ is Gaussian, and by integrating (7) it can be shown that its limiting variance agrees with the expression in [25].

## 3.2 Max-sliced Wasserstein distance

Unlike integration, the supremum functional is not smooth and does not possess a linear Hadamard derivative. Write $MSW_p^p(P,Q) = \sup_{u \in \mathbb{S}^{d-1}} W_p^p(P_u, Q_u)$, and note that $MSW_p^p(P,Q) = \omega(W)$, where $\omega : \ell^\infty(\mathbb{S}^{d-1}) \to \mathbb{R}$ is the supremum functional. It is shown in Theorem 2.1 of [5] that $\omega$ is Hadamard directionally differentiable with the derivative

$$\omega'_f(g) = \lim_{\epsilon \downarrow 0} \sup_{x \in A_\epsilon(f)} g(x),$$

where $A_\epsilon(f) := \{x : f(x) \geq \sup f - \epsilon\}$. Moreover, if $f$ and $g$ are continuous on $\mathbb{S}^{d-1}$ with respect to the standard Euclidean distance, then $\lim_{\epsilon \downarrow 0} \sup_{A_\epsilon(f)} g(x) = \sup_{x \in A_0(f)} g(x)$ where $A_0(f) = \{x : f(x) = \sup f\}$. (See Corollary 2.3 of [5])

Applying the functional delta method to $\omega$ and the uniform weak convergence (6), we note that the limiting process $\mathbb{G}$ has continuous samples paths a.s., so the limiting distribution of MSW can be written as

$$\omega'_{W_p^p(W)}(\mathbb{G}) = \sup_{u \in A_0(W_p^p(W))} \mathbb{G}(u). \tag{10}$$

In the spiked transport model (STM), this expression can be further simplified. The STM was introduced by [29] to formalize the situation where two distributions differ only in a low dimensional subspace of $\mathbb{R}^d$. We describe the special case of one-dimensional spike here. Fix some $v \in \mathbb{S}^{d-1}$ and let $X, Y \in L := \mathrm{span}(v)$ be two random variables with different laws. Let $Z$ be another random variable independent of $(X, Y)$ and supported on the orthogonal complement $L^\perp$ of $L$.

Then we define two distributions in $\mathbb{R}^d$ by $P := \text{law}(X + Z)$, $Q := \text{law}(Y + Z)$. It is shown there that $MSW_p^p(P, Q) = W_p(\text{law}(X), \text{law}(Y)) = W_2(P, Q)$. In addition, in this model the set $A_\epsilon(W_p^p(P., Q.))$ shrinks to the singleton set $\{v\}$ as $\epsilon$ goes down to $0$. In fact, the Hadamard derivative can be reduced to the random variable $\mathbb{G}(u)$, i.e. the marginal of the limiting Gaussian process $\mathbb{G}$ along $v$. We summarize the result in the following theorem.

**Theorem 3.2.** *Suppose that two compactly supported probability distributions $P$ and $Q$ in $\mathbb{R}^d$ fit the spiked transport model with spike $v \in \mathbb{S}^{d-1}$. Assume furthermore $P$ and $Q$ also satisfy (CC). Then,*

$$\sqrt{n}\left(MSW_{p,1}^p(P_n, Q_n) - MSW_{p,1}^p(P, Q)\right) \xrightarrow{d} \mathbb{G}(v). \tag{11}$$

**Remark 3.3.** *Note that it is not necessary for $P$ and $Q$ to satisfy the spiked transport model in order to deduce the CLT of max-sliced Wasserstein. Namely, even if the set $\{u \in \mathbb{S}^{d-1} : W_p^p(P_u, Q_u) = MSW(P, Q)\}$ is not a singleton, the same proof still works but the limiting distribution is the supremum of the Gaussian process $\mathbb{G}$ over the set $\{u \in \mathbb{S}^{d-1} : W_p^p(P_u, Q_u) = MSW(P, Q)\}$ which is not necessarily Gaussian. This example shows that certain functionals give rise to non-Gaussian limits, even though the limit in (6) is Gaussian. We given an example of this behavior in the supplementary material.*

### 3.3 Distributional Sliced Wasserstein Distance

Proposed in [28], the distributional sliced Wasserstein distance is a generalization of sliced Wasserstein distance. Formally, given two probability measures $P$ and $Q$ on $\mathbb{R}^d$ with finite $p$-th moments where $p > 1$ and a subset of probability distributions $\mathcal{P}_C$ on $\mathbb{S}^{d-1}$ such that $\mathbb{E}_{\theta,\theta' \sim \tau}|\theta^\top \theta'| \leq C$ for all $\tau \in \mathcal{P}_C$ for some constant $C > 0$, the distributional sliced $p$-sliced Wasserstein distance between $P$ and $Q$ is defined by

$$DSW_p(P, Q; C) := \sup_{\tau \in \mathcal{P}_C} \left(\int_{\mathbb{S}^{d-1}} W_p^p(P_\theta, Q_\theta) \, \mathrm{d}\tau(\theta)\right)^{1/p}.$$

We may therefore write $DSW_p^p(P, Q; C) = \omega_C(F_C(W))$, where $\omega_C : \ell^\infty(\mathcal{P}_C) \to \mathbb{R}$ is the supremum functional and $F_C : \ell^\infty(\mathbb{S}^{d-1} \to \mathcal{P}_C)$ is defined by $F_C(\Phi)(\cdot) := \int_{\mathbb{S}^{d-1}} \Phi(u) \, \mathrm{d} \cdot (u)$.

The function $F_C$ is trivially Hadamard differentiable following the same argument for the standard sliced Wasserstein distance given above. The supremum functional on $\ell^\infty(\mathcal{P}_C)$ is also Hadamard directionally differentiable by Theorem 2.1 of [5]. Since the composition of Hadamard differentiable functions are still Hadamard directionally differentiable (see e.g. [2, Proposition 2.47]), under the assumption (CC), may conclude that $\omega_C \circ F_C$ is Hadamard directionally differentiable, i.e.

$$\sqrt{n}\left(DSW_p^p(P_n, Q_n; C) - DSW_p^p(P, Q; C)\right) \xrightarrow{d} \lim_{\epsilon \downarrow 0} \sup_{\tau \in A_\epsilon(DSW_p^p(P,Q;C))} \int_{\mathbb{S}^{d-1}} \mathbb{G}(\theta) \, \mathrm{d}\tau(\theta). \tag{12}$$

## 4 Simulation Studies

We illustrate our distributional limit results in Monte Carlo simulations. Specifically, we investigate the speed of convergence of the sliced Wasserstein distance and the max-sliced Wasserstein distance. We also investigate the convergence speed of the *amplitude*, which provides an example of a functional not covered in prior work.

We also illustrate the accuracy of the approximation using the re-scaled bootstrap. All simulations were performed using Python. The Wasserstein distances as well as the sliced Wasserstein distances were calculated using the Python package *POT* [18] and the max-sliced Wasserstein distances were approximated by the Riemannian optimization method proposed in [23].

### 4.1 Sliced Wasserstein Distance

We present an example that concerns two different distributions with connected projections along all directions. Consider a simple model of transport where source and target distributions $P$, $Q$ are uniform on unit sphere $\mathbb{S}^2$ and the unit sphere $\mathbb{S}^2_{(1,1,1)}$ centered at $(1, 1, 1)$ respectively.

We first give an explicit representation of the theoretical limit of the example given in section 3.1. Fix any point $\theta \in \mathbb{S}^2$, the projections of $P$ and $Q$ along $\theta$ are uniform over $(-1, 1)$ and $(-1 + a_\theta, 1 + a_\theta)$ respectively where $a_\theta := \theta_1 + \theta_2 + \theta_3$. Then the unique Kantorovich potential that achieves 2-Wasserstein distance between $P_\theta$ and $Q_\theta$ is $\phi_0^\theta(x) = -2a_\theta x$. Hence, we have

$$\sqrt{n}\left(W_2^2(P_{n\cdot}, Q_{n\cdot}) - W_2^2(P_\cdot, Q_\cdot)\right) \rightsquigarrow \mathbb{G},$$

where $\mathbb{G}$ is the mean-zero Gaussian process indexed by $\mathbb{S}^2$ with covariance functions

$$\mathbb{E}\mathbb{G}(u)\mathbb{G}(v) = \frac{8}{3}a_u a_v \langle u, v \rangle.$$

It follows from Theorem 3.1 that the limiting distribution of the empirical 2-Wasserstein distance is the centered Gaussian $S$ with variance

$$\text{Var}(S) = \frac{8}{3} \int_{\mathbb{S}^2} \int_{\mathbb{S}^2} a_u a_v \langle u, v \rangle \, d\sigma(u) d\sigma(v) \approx 0.832.$$

We sample i.i.d. observations $X_1, \ldots, X_n \sim P$ and $Y_1, \ldots, Y_n \sim Q$ with size $n = 50, 100, 500$. This process is repeated 500 times. We then compare the finite distributions of 1-Wasserstein distance with the theoretical limit given in section 3.1. We demonstrate the results using kernel density estimators in Figure 1 along with the corresponding Q-Q plots. We see that the finite-sample empirical distribution gets closer to the limiting Gaussian distribution in (9) as the sample size $n$ increases. In addition, we simulate the re-scaled plug-in bootstrap approximations by sampling $n = 1000$ observations of $P$ and $Q$. Fix some empirical SW $\sqrt{n}SW_2^2(P_n, Q_n)$, we generate $B = 500$ replications of $\sqrt{l}(SW_2^2(\hat{P}_n^*, \hat{Q}_n^*) - SW_2^2(P_n, Q_n))$. The distributions of the replications with various replacement numbers $l$, compared with the finite-sample empirical distribution and the theoretical limit, are shown in Figure 2. We observe that the naive bootstrap ($l = n$) better approximates the finite sample distribution compared to fewer replacements ($l = n^{1/2}, n^{3/4}$). This is consistent with the observation of inference on finite spaces. [30]

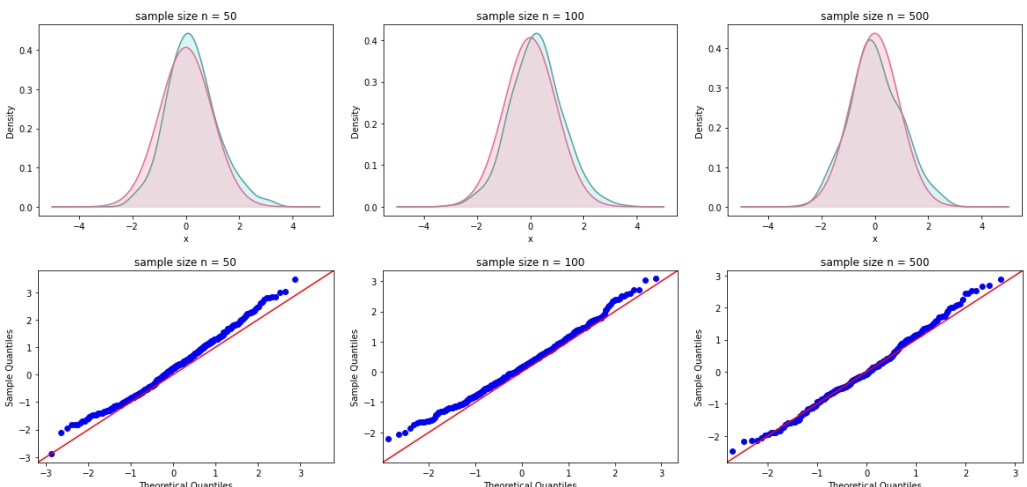

Figure 1: Top: Comparison of the finite sample density (pale turquoise) and the limit distribution of the empirical sliced distance (pink).
Bottom: The corresponding Q-Q plots where the red solid line indicates perfect fit.

## 4.2 Max-sliced Wasserstein Distance

We present an example that simulates the behavior of the max-sliced Wasserstein distance when $p = 2$. We take $P$ to be the uniform distribution on the unit sphere $\mathbb{S}^2$ and $Q$ to be uniform on the surface of ellipsoid $x^2/a^2 + y^2 + z^2 = 1$ where $a = 8.5$. We sample i.i.d. observations with size $n = 50, 100, 500$ and this process is repeated 2000 times. The estimation plotted in the top part of

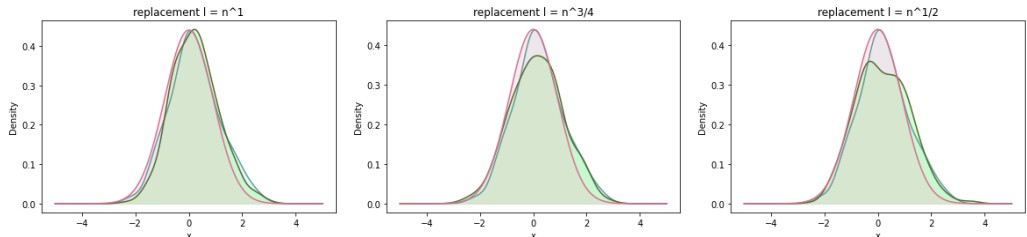

Figure 2: Bootstrap for the empirical sliced distance. Illustration of the re-scaled plug-in bootstrap approximation ($n = 1000$) with replacement $l \in \{n,, n^{3/4}, n^{1/2}\}$. Finite bootstrap densities (pale green) are compared to the corresponding finite sample density (pale turquoise) and the limit distribution (pink).

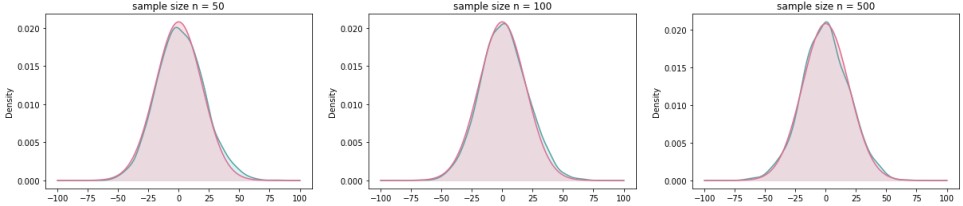

Figure 3: Comparison of the finite sample density (pale turquoise) and the limit distribution of the empirical max-sliced Wasserstein distance (pink).

Figure 3 indicates that the finite sample distributions approximate the limiting Gaussian distribution derived in Theorem 3.2 very well even when the sample size is small.

In terms of the re-scaled bootstrap, the accuracy of the bootstrap approximation seems to be good for all replacement numbers in this case, which is again consistent with the observation of the case when the underlying distributions are supported in finite sets. [30] See Figure 4 for the simulation.

## 4.3 Amplitude

In this section, we give an example of a new functional, the amplitude, to which our theory applies. For $f \in \ell^\infty(\mathbb{S}^{d-1})$, we write $\mathrm{amp}(f) := \sup f - \inf f$. When $Q$ is chosen to be a radially symmetric reference distribution, e.g., uniform on the sphere, the quantity

$$\mathrm{amp}(W_2^2(P_\cdot, Q_\cdot)) = \sup_{u \in \mathbb{S}^{d-1}} W_2^2(P_u, Q_u) - \inf_{u \in \mathbb{S}^{d-1}} W_2^2(P_u, Q_u)$$

is the natural measure of the radial homogeneity of $P$—if $\mathrm{amp}(W_2^2(P_\cdot, Q_\cdot))$ is small, then $P$ differs from $Q$ by similar amounts in each direction. The amplitude functional defined on $\ell^\infty(\mathbb{S}^2)$ is

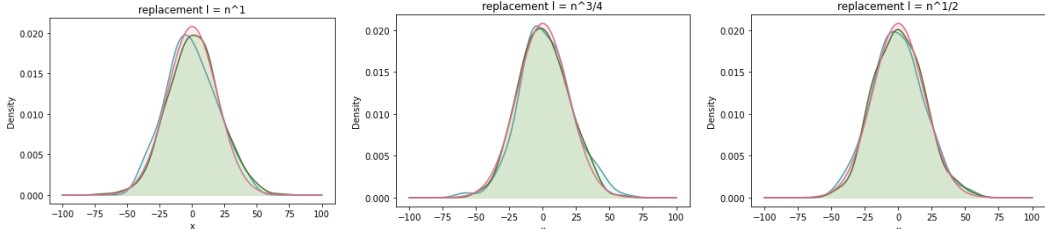

Figure 4: Bootstrap for the empirical max-sliced Wasserstein distance. Illustration of the re-scaled plug-in bootstrap approximation ($n = 1000$) with replacement $l \in \{n, n^{3/4}, n^{1/2}\}$. Finite bootstrap densities (pale green) are compared to the corresponding finite sample density (pale turquoise) and the limit distribution (pink).

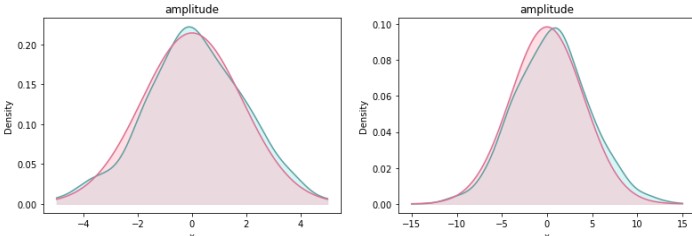

Figure 5: Comparison of the finite sample density (pale turquoise) and the limit distribution of the empirical PRW (pink).
Left: $\mathcal{P} \sim U(\{\frac{x^2}{4} + 4y^2 + z^2 = 1\})$, Right: $\mathcal{P} \sim U(\{\frac{x^2}{4} + \frac{y^2}{4} + \frac{z^2}{16} = 1\})$. $\mathcal{Q} \sim U(\mathbb{S}^2)$.

Hadamard directionally differentiable. [5]

Consider $P$ being uniform over the surface of the ellipsoid $\{x^2/4 + 4y^2 + z^2 = 1\}$ and $Q$ uniform on $\mathbb{S}^2$. Applying Corollary 2.4 to $P, Q$ and amp, we obtain

$$\sqrt{n}\left(\text{amp}(W_2^2(P_{n\cdot}, Q_{n\cdot})) - 5/4\right) \xrightarrow{d} \mathbb{G}((1, 0, 0)).$$

We simulate the density of the amplitude of empirical Wasserstein distances of 1d projections. The finite sample density generated by $n = 600$ samples and theoretical limit are given in Figure 5.

Let $P$ be uniform on $\{x^2/4 + y^2/4 + z^2/16 = 1\}$ and keep $Q$ unchanged. Then

$$\sqrt{n}\left(\text{amp}(W_2^2(P_{n\cdot}, Q_{n\cdot})) - 8/3\right) \xrightarrow{d} \mathbb{G}((0, 0, 1)) - \mathbb{G}((0, 1, 0)).$$

We generate $n = 600$ samples according to $P$ and $Q$ and the result is also shown in Figure 5.

Both finite sample densities indeed converge to the theoretical Gaussian limits as the sample size increases.

## 5 Conclusion

This paper defines the *Sliced Wasserstein Process*, a stochastic process indexed by elements of the unit sphere $\mathbb{S}^{d-1}$ in $\mathbb{R}^d$, and shows that under regularity assumptions on $P$ and $Q$, this process converges to a tight Gaussian process on $\ell^\infty(\mathbb{S}^{d-1})$. This convergence result, which can be viewed as a uniform central limit theorem for the empirical Wasserstein distance along all directions simultaneously, immediately implies distributional convergence and bootstrap consistency results for the sliced Wasserstein distance and its variants, thereby unifying and streamlining existing proofs in the literature and providing distributional limits for variants of the sliced Wasserstein distance for which no such results were previously known.

An important question left open by our work is whether a similar result holds under weaker assumptions on $P$ and $Q$. We conjecture that the compact support assumption can be lifted, though doing so would likely require making relatively stringent tail conditions. Avoiding assumption (**CC**) is more subtle, as some assumption of uniqueness of potentials is required to obtain Gaussian limits.

Finally, we anticipate that our techniques can also be applied to entropically regularized variants of the Wasserstein distance, where empirical process theory arguments have also been central in proving both sample complexity and distributional limit results. We leave this extension to future work.

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
