# OpenReview forum: "Distributional Convergence of the Sliced Wasserstein Process"
_NeurIPS.cc/2022/Conference — NeurIPS 2022 Accept_

### Official Review · Reviewer_84Jv · 2022-07-06

**Rating:** 5
**Confidence:** 4
**Soundness:** 3 good
**Presentation:** 2 fair
**Contribution:** 2 fair

**Summary:**

This paper defines the Sliced Wasserstein process, a stochastic process defined by the empirical Wasserstein distance between slices of empirical probability measures. A uniform distributional limit theorem is derived. Experimental results demonstrate the convergence towards the limiting Gaussian process.

**Questions:**

- A question I have regarding the SW process is the following: G_n defined in Equation 2 is said to enjoy a uniform central limit theorem. My naive question is: given the fact that $\mathbb{E}[W_p^p(P_{nu}, Q_{nu})]\neq W_p^p(P_u,Q_u)$ (due to the permutation/transport plan of the Wasserstein distance), then G_n is defined according to the "wrong expectation"? (see Corollary 2 of https://proceedings.neurips.cc/paper/2020/file/eefc9e10ebdc4a2333b42b2dbb8f27b6-Paper.pdf for instance where you need to compute a sample complexity). Maybe what I am saying is not true because the authors are considering a single slice. I would like to have their opinion on this.

- How hard would it be to go beyond the compact case? At least Sub-Gaussian.

- I suggest the authors should use the remaining space they have (1.5 pages!) to develop sketches of proof in the main paper, or do more experiments, maybe to show that their theory is valid for the unbounded case numerically?

I am willing to raise my score accordingly if the answers to my questions are satisfying and the weaknesses addressed.

**Limitations:**

No potential negative impact

**Strengths And Weaknesses:**

Strengths:
- This work build upon the mathematical techniques developed in [20] and adapt them well for the SW process.
- The results seem correct (I just skimmed through the proofs).

Neutral:
- Numerical results are convincing but I would have expected more examples, maybe ones that show the theory also hold (or not?) in the unbounded case.

Weaknesses:
- The paper is not very clear: more space could have been given for defining mathematical objects; a sketch of proof of the main theorem could have been given knowing that there is a lot of space not used at the end of the paper.
- Contrary to what is said in the Checklist, the code is not given (it is specified it is given in a footnote shown page 6), and I would have liked to look at it.
- The authors limit themselves to compactly supported probability measures P and Q (I am not certain if it’s possible to go beyond compact, maybe sub-Gaussian?). At least the authors could have developed a bit on why they considered this assumption and where are the difficulty for the unbounded case.
- The authors did not verified the documents they sent: all the labels of equations and sections are visible…
- There are a lot of typos (for instance the title of the SM does not correspond to the paper's title).
- I don't see any real applications for these results: I know NeurIPS accepts very theoretical results, but it would be useful to know how this could improve or help us understand SW distances better given the already known results on the theory of SW.

---

> ### Author Response · Authors · 2022-08-02
> **Author Response**
>
> We thank the reviewer for the thorough reading and thoughtful comments. We believe the reviewers' comments have helped us strengthen the draft considerably, and we invite the reviewer to reconsider their recommendation in light of our revision.
>
> 1. Centering at expectation:
>
> We appreciate the reviewer raising this important point.
>
> For the purposes of inference in practice, it is much better to obtain a limit result centered at $W_p^p(P_u, Q_u)$ (which is the population-level quantity of interest) rather than at $\mathrm{E} W_p^p(P_{nu}, Q_{nu})$ (which does not possess a practical interpretation in terms of the measures $P$ and $Q$).
>
> However, most of the distributional limit results in the OT literature are centered at the latter quantity, and developing results centered at the former, population-level quantity often requires significant effort.
> (For a further discussion of the difficulty of obtaining limits with the correct centering, see the introduction to ref. [21] in the original submission.)
>
> The fact that our process is centered at $W_p^P(P_u, Q_u)$ is therefore an important benefit of our results and approach, and we have emphasized this fact in our revision.
>
> To connect this with the results obtained in the referenced paper of Nadjahi et al., we note that their Corollary 2 is $\textit{not}$ sharp under assumption $\textbf{(CC)}$ which holds throughout out work, and in fact under this assumption $|\mathrm{E} SW_p^p(P_n, Q_n) - SW_p^p(P, Q)| = o(n^{-1/2})$ for $p > 1$.
>
> 2. Possible extension to a more general setting where P and Q are subgaussian:
>
> The assumption of compact support is indeed restrictive.
> However, it represents a relatively common state of affairs in optimal transport, where theorems are first established in the compactly supported case and only later extended to the unbounded setting, often with different techniques.
> Obtaining this extension for the sliced Wasserstein process is an attractive question for future work.
>
>  The compactness assumption is used to guarantee that the set of Kantorovich potentials corresponding to $P_u$ and $Q_u$ for any $u \in \mathbb{S}^{d-1}$ is uniformly Lipschitz, and is therefore a subset of a Donsker class. If the supports of $P$ and $Q$ were unbounded, in order to deduce the Donsker property, we would need additional assumptions on the $1$-dimensional projections of $P$ and $Q$ as well as the cost function, (see e.g. Theorem 5.2 of [hundrieser2022unifying]) that do not hold for $p$-Wasserstein distances ($p > 1$) in general. We have added a remark clarifying this point after Theorem 2.1.
>
> 3. Code:
>
> In a late stage of editing, we removed a link to a Github version of the code due to anonymity concerns. We have uploaded an anonymized version with the revised supplement and apologize for creating confusion.
>
> 4. Remaining space and small typos:
>
> We thank the reviewer for pointing out the typos and the problem of remaining space. We have used the additional space to offer another application of our techniques (to the distributional sliced Wasserstein distance) and further discussion on the implications of our results throughout.
>
> [hundrieser2022unifying] Marcel Klatt and Carla Tameling and Axel Munk, Empirical Regularized Optimal Transport: Statistical Theory and Applications, SIAM J. Math. Data Sci., 2, 419-443, 2020.

---

> > ### Comment · Reviewer_84Jv · 2022-08-07
> > **Answer to the authors**
> >
> > I would like to thanks the authors for their answer to my questions.
> >
> > I especially appreciate the answer about the centering in expectation, where they explain that their method is naturally centered in $W_p^p(P_u,Q_u)$. I however truly think that having the Subgaussian case would have strengthened the paper much more and make it more complete.
> >
> > I therefore decide to raise my score by 1.

---

### Official Review · Reviewer_2q9t · 2022-07-07

**Rating:** 6
**Confidence:** 4
**Soundness:** 3 good
**Presentation:** 3 good
**Contribution:** 2 fair

**Summary:**

The authors present a unified method for providing distributional limits for processes constructed from the empirical Wasserstein distance between one dimensional projections of empirical distributions. To do so, they rely on the theory of empirical processes and on the Hadamard directional differentiability of various functions.

**Questions:**

- Can you specify where the assumption of compactness of the support of the probability measure in $\mathbb{R}^d$ is necessary?

- As proved in [Nadjahi], the sliced-Sinkhorn divergences do not depend on the entropy regularization parameter, unlike the entropy regularized optimal transport (or Sinkhorn divergence). Therefore, considering sliced-Sinkhorn divergences could also be of interest. As similar empicial process methods have been used for entropy regularized OT, would it be possible to extend you result to this particular framework?

- It could be made clear from the beginning that the proposed method is in any case limited to absolutely continuous probability measures, as their support must be connected.

- Since your results apply for various functions F, it might have been interesting to propose a new divergence to assert the interest of your generalisation. I however acknowledge that it is not an easy task to propose a new satisfactory distance.

- More could be said about the trimmed sliced Wasserstein distance, and the precise meaning of "robustification".

- In the appendix B.1, you consider the case $p=1$, which is not included in the Wasserstein distance setting that you develop (Section 2). In particular Theorem 2.1 is stated for the case $p>1$. Are the results of appendix B.1 can be obtain directly? If so, this could be clarified.

[Nadjahi] - "Statistical and Topological Properties of Sliced Probability Divergences", Nadjahi et al.

**Limitations:**

The authors have specified that their method is quite restrictive in the sense that it only applies to a rather small set of probability distributions.

**Strengths And Weaknesses:**

This article follows a natural set of results that have been proven for both classical and entropic optimal transport. The presentation is clear and the article is easy to read. I develop the strengths and weaknesses further in the following.

Strengths:
- The paper proposes a unified method to prove distributional limits for distances based on sliced distributions and 1D Wasserstein distance. The method is based on a previous work [20], that addressed the case of empirical optimal transport.

Weaknesses:
- The novelty of this article seems to be limited as most of the major results have been proven in [19,23].
- The assumption (CC) that the support of the projected measures is connected for all the vector in the unit sphere is very restrictive. I agree that the general case involves a complicated geometry of the distributions, however, the unified framework on which the paper is based concerns very few applied cases.
- The method for proving the results of the paper uses classical tools of weak convergence of empirical processes and does not particularly differ from the case of classical optimal transport.

Overall, the results presented here are restricted to probability measures under strict assumptions and are therefore rather limited: every one dimensional projection of probability distributions onto the unit sphere must have an interval as a support. Moreover, this paper seems to have been published in a hurry (see "Minor comments" below), there is notably no conclusion and no perspective of using such results for example.

Minor comments:
- The labels are visible in the margin.
- The reference [28,29] appears twice.
- l.94 : "uniquness"
- In the proof of Theorem 2.1 and in Corollary 2.3, the random variables $X_i$ and $Y_i$ are not defined.
- In section 3.2, it is question of MSW, $\tilde{W}$ and WPP (which is not defined). Standardisation of terminology is necessary.
- In Theorem 3.1, the dependence on $\delta$ of the trimmed SW disappeared.
- l. 190 : "given in section 4.1" should be removed/rewritten.

---

> ### Author Response · Authors · 2022-08-02
> **Author Response**
>
> We thank the reviewer for the thorough reading and thoughtful comments.
>
> 1. Novelty of results:
>
> The reviewer argues that our main results appear already in references [20, 24]. To restate our goal with this paper: we aim to define a new object (the sliced Wasserstein process) whose convergence explains, and yields as a corollary, the results of [20] and [24]. We view this as an important conceptual step forward in the understanding of the sliced Wasserstein distance and its variants, and allows direct derivation of limit results for variants for which no limit results currently exist. For example, we are aware of no limit theorems for the Distributional Sliced Wasserstein distance defined in [27]. Corollaries 2.2 and 2.3 immediately imply a central limit theorem and bootstrap consistency result for this distance, which we have added as an additional example to the paper. We believe the ease with which these and other theorems can be derived from our main result illustrates the impact of our approach.
>
> 2. Necessity of the assumption of compactness of the supports of the probability measures:
>
> The assumption of compact support is indeed restrictive.
> However, it represents a relatively common state of affairs in optimal transport, where theorems are first established in the compactly supported case and only later extended to the unbounded setting, often with different techniques.
> Obtaining this extension for the sliced Wasserstein process is an attractive question for future work.
>
> The compactness assumption is used to guarantee that the set of Kantorovich potentials corresponding to $P_u$ and $Q_u$ for any $u \in \mathbb{S}^{d-1}$ is uniformly Lipschitz, and is therefore a subset of a Donsker class. If the supports of $P$ and $Q$ were unbounded, in order to deduce the Donsker property, we would need additional assumptions on the $1$-dimensional projections of $P$ and $Q$ as well as the cost function, (see e.g. Theorem 5.2 of [hundrieser2022unifying]) that do not hold for $p$-Wasserstein distances ($p > 1$) in general. We have added a remark clarifying this point after Theorem 2.1.
>
> 3. State the assumption as absolutely continuous probability measures instead of bounded connected supports:
>
> We respectfully disagree with the reviewer's assertion that our results are limited to absolutely continuous probability measures.
> For instance, there exist probability measures that are not absolutely continuous with respect to the Lebesgue measure on $\mathbb{R}^d$ but whose one-dimensional projections have support which is an interval (e.g., the uniform measure over the unit sphere in $\mathbb{R}^d$).
> The one-dimensional projections need not even be absolutely continuous; for instance, consider the distribution on $[0, 1]$ whose CDF is $F(x) = x/2$ for $x \leq 1/2$ and $F(x) = x/2 + 1/2$ for $x > 1/2$.
> In short, though the assumption \textbf{(CC)} does exclude some examples, it does not restrict the application of the theorem to absolutely continuous measures.
>
> 4. Trimmed sliced Wasserstein distance:
>
> We thank the reviewer for pointing out that this was not clearly explained in our original submission. We have borrowed this terminology from [24], and have added more context to the revision.
>
> 5. Extension to sliced-Sinkhorn divergences:
>
> Extending our framework to the case of entropy-regularized transport is indeed possible and relatively straightforward, to obtain a "sliced Sinkhorn process". We have not pursued this direction in the interest of keeping focus on the un-regularized sliced distances (which are more common in practice), but agree that the extension to the regularized case is interesting.
>
> 6. Example corresponding to $p = 1$ in Appendix B.1:
>
> This is a good point that we should have clarified. The only obstacle to include $p = 1$ in the main result is that there do not seem to be general conditions under which the Kantorovich potentials under $p = 1$ are unique. The assumption $(CC)$ only works for $p > 1$. In the simulation we present in Appendix B.1, we have verified directly that the potentials corresponding to all one-dimensional projections are unique, so that the main distributional convergence result still holds. We will clarify this.
>
> [hundrieser2022unifying] Marcel Klatt and Carla Tameling and Axel Munk, Empirical Regularized Optimal Transport: Statistical Theory and Applications, SIAM J. Math. Data Sci., 2, 419-443, 2020.

---

> > ### Comment · Reviewer_2q9t · 2022-08-08
> > **Response to authors**
> >
> > Thank you very much for your detailed response and the changes made to the document (especially the clarification for the example with $p=1$). Also, I agree with you on the particular issue raised in 2, thanks for pointing out my misunderstanding.
> >
> > After reading the other reviews and responses, the authors' goal of proposing a general strategy to study the limiting distributions of the sliced Wasserstein processed was made more clear.
> >
> > I have therefore increased my score.

---

### Official Review · Reviewer_mUtM · 2022-07-10

**Rating:** 6
**Confidence:** 4
**Soundness:** 3 good
**Presentation:** 2 fair
**Contribution:** 3 good

**Summary:**

Optimal transport distances (a.k.a. Wasserstein distance)  have recently drawn ample attention in statistics and machine learning communities as powerful discrepancy measures for probability distributions. One of the bottlenecks of Wasserstein distance consists of its expensive computational cost. An alternative to making this problem computationally tractable leads in the fact that it can be cast on averaging calculations of $1$-D Wasserstein distances. This latter approach is referred to as Sliced Wasserstein distance (SWD). In a nutshell, SWD relies on projecting the data sample on some $u$-direction from the unit sphere, ($u \in \mathbb{S}^{d-1}).$

This paper addresses a theoretical unified problem of proving a Central Limit Theorem for the empirical SWD distance under a compact supports condition satisfied by the distributions in question.  The authors show that  the asymptotic distribution is a tight-centered Gaussian process on the unit sphere $\mathbb{S}^{d-1}$.



**Questions:**


- For the discrete SWD:
    - can one get directly the central limit from the one verified by the Wasserstein distance?
    - does this limit process uniform with respect to $L$, the number of $u$-random direction projections. Namely, if there is $L \leq L'$ random projections, the limit is (approximately) the same?
- In Theorem 2.4., can one derive an order of the convergence rate to zero?

**Limitations:**

This is not applicable to the paper.

**Strengths And Weaknesses:**

The paper is easy to follow, and it presents a unified framework for proving central limit theorem for a family of SWD distance, including vanilla SWD, Max-Sliced, Trimmed SWD, etc. This result in itself is solid, especially for statistical applications involving SWD.

---

> ### Author Response · Authors · 2022-08-02
> **Author Response**
>
> We thank the reviewer for the thorough reading and thoughtful comments.
>
> 1. Discrete SWD:
>
> The fact that the discrete SWD enjoys a Gaussian limit can indeed be deduced from the CLT for the Wasserstein distance, though this CLT for the Wasserstein distance does not directly imply what the limiting variance for the discrete SWD will be. Obtaining the limiting variance requires a joint convergence result like Theorem 2.1.
>
> There is stability in the limiting distribution for various values of $L$. Examining the form of the limiting covariance in Eq.~(7), conditioned on the (potentially random) directions $u_1, \dots, u_L$, we obtain that $\sqrt{n}(\widehat{SW}_p^p(P_n, Q_n) - \widehat{SW}_p^p(P, Q)) \rightsquigarrow \mathcal{N}(0, \sigma^2)$ where
>
> $\sigma^2 = (1-\lambda)\mathrm{var}(P)(\frac{1}{L}\sum_{i=1}^L f_{u_i}(u_i^\top X)) + \lambda \mathrm{var}(Q)(\frac{1}{L}\sum_{i=1}^L f^c_{u_i}(u_i^\top Y))$.
>
> As $L \to \infty$, if the $u_i$ are drawn independently at random from the uniform distribution on the sphere, then this quantity converges almost surely to the limiting variance corresponding to the standard SWD.
> In particular, for $L, L'$ sufficiently large, the limiting variances will be close with high probability.
>
> 2. Rate of convergence in Theorem 2.4:
>
> We presented Theorem 2.4 to permit the construction of asymptotic confidence intervals. By Slutsky's theorem, a consistency result (e.g., Theorem 2.4) suffices for this purpose, and we have clarified this point in our revision. Since our focus was on asymptotic inference, we did not pursue a quantitative version of this result, but we believe it to be a quite interesting question for future work. Such a result which would necessitate the development of quantitative convergence bounds for Kantorovich potentials, which to our knowledge are not known.

---

### Official Review · Reviewer_enEA · 2022-07-11

**Rating:** 7
**Confidence:** 3
**Soundness:** 4 excellent
**Presentation:** 4 excellent
**Contribution:** 3 good

**Summary:**

The paper provides a Donsker-typed theorem for Projected Wasserstein distance in $l^{\infty}$ norm over the projected directions ball $S^{d-1}$. From that, it derives Limit Theorems for some well-known Sliced Wasserstein distances. Some simulation studies are carried out to support the theoretical findings.

**Questions:**

1. Throughout the paper, the distributions of interest $P$ and $Q$ are assumed to be compactly supported. A natural question is "can we extend the result to a more general setting where $P$ and $Q$ have arbitrary supports?". It seems like the compact condition of the supports is only needed to show the existence of the potential and to bound the entropy number. If we can extract those refined conditions and put them together into a more general result (maybe in Appendix), it would be more impactful to OT research. If it is not possible, it is helpful to have a paragraph to explain why.

2. It would be interesting to see some simulation studies about the limiting distribution of the empirical process of Max-Sliced Wasserstein distances, even in low-dimensional cases. It would be helpful to build some intuition about this non-Gaussian limit.

**Ethics Review Area:**

["I don’t know"]

**Limitations:**

The paper is well written and has no major limitations.

**Strengths And Weaknesses:**

**Strength**:

1. The results in this paper are important for Optimal Transport theory and application. The presentation of the paper is clean and easy to read.

2. All the proofs are solid and written carefully.

---

> ### Author Response · Authors · 2022-08-02
> **Author Response**
>
> We thank the reviewer for the thorough reading and thoughtful comments.
>
> 1. Possible extension to a more general setting where P and Q have arbitrary supports:
>
> The assumption of compact support is indeed restrictive.
> However, it represents a relatively common state of affairs in optimal transport, where theorems are first established in the compactly supported case and only later extended to the unbounded setting, often with different techniques.
> Obtaining this extension for the sliced Wasserstein process is an attractive question for future work.
>
> As the reviewer notes, the compactness assumption is used to guarantee that the set of Kantorovich potentials corresponding to $P_u$ and $Q_u$ for any $u \in \mathbb{S}^{d-1}$ is uniformly Lipschitz, and is therefore a subset of a Donsker class. If the supports of $P$ and $Q$ were unbounded, in order to deduce the Donsker property, we would need additional assumptions on the $1$-dimensional projections of $P$ and $Q$ as well as the cost function, (see e.g. Theorem 5.2 of [hundrieser2022unifying]) that do not hold for $p$-Wasserstein distances ($p > 1$) in general. We have added a remark clarifying this point after Theorem 2.1.
>
> 2. Simulation studies about the Max-sliced Wasserstein distance:
>
> We included an example of the max-sliced Wasserstein distance in section 4.2 where we call it ``Wasserstein project pursuit,'' which is alternate terminology (proposed by ref. [28]) for the max-sliced Wasserstein distance. We have modified this section to clarify that this section illustrates the limit for the max-sliced Wasserstein distance. We have also added an example with a non-Gaussian limit in section B.2 of the appendix.
>
> [hundrieser2022unifying] Marcel Klatt and Carla Tameling and Axel Munk, Empirical Regularized Optimal Transport: Statistical Theory and Applications, SIAM J. Math. Data Sci., 2, 419-443, 2020.

---

> > ### Comment · Reviewer_enEA · 2022-08-05
> > **Reviewer response**
> >
> > Thank you for providing answers to my questions. I understood the theoretical difficulties that the authors have for the non-compact support distributions and I believe that the available technique may not resolve it easily. I think the paper is a good contribution to the statistical optimal transport theory community.

---

### Meta-Review · Area_Chair_H47x · 2022-08-26

**Recommendation:** Accept
**Confidence:** Certain

**Metareview:**

After the rebuttal period, the reviewers have come to an agreement on the paper being novel, interesting, the contributions being significant. The rebuttal also addressed most of the concerns, though I agree with reviewer 84Jv on the comment that experiments on non-compact settings would be a plus to see the limits of the theory. Overall, I believe this is a nice continuation for the prior art and I recommend an acceptance for the paper.

**Award:**

No

---

### Decision · Program_Chairs · 2022-09-14

Accept